# Circulating Metabolites as Biomarkers of Disease in Patients with Mesial Temporal Lobe Epilepsy

**DOI:** 10.3390/metabo12050446

**Published:** 2022-05-17

**Authors:** Alexandre B. Godoi, Amanda M. do Canto, Amanda Donatti, Douglas C. Rosa, Danielle C. F. Bruno, Marina K. Alvim, Clarissa L. Yasuda, Lucas G. Martins, Melissa Quintero, Ljubica Tasic, Fernando Cendes, Iscia Lopes-Cendes

**Affiliations:** 1Department of Translational Medicine, School of Medical Sciences, University of Campinas (UNICAMP), Campinas 13083-888, Brazil; bdgalexandre@gmail.com (A.B.G.); amanda.morato.canto@gmail.com (A.M.d.C.); donatti.amanda@gmail.com (A.D.); douglascescon@gmail.com (D.C.R.); danielle.carmof@gmail.com (D.C.F.B.); 2Brazilian Institute of Neuroscience and Neurotechnology (BRAINN), Campinas 13083-888, Brazil; marinakma@gmail.com (M.K.A.); cyasuda@unicamp.br (C.L.Y.); fcendes@unicamp.br (F.C.); 3Department of Neurology, School of Medical Sciences, University of Campinas (UNICAMP), Campinas 13083-888, Brazil; 4Department of Organic Chemistry, Institute of Chemistry, University of Campinas (UNICAMP), Campinas 13083-888, Brazil; lgmartins1984@gmail.com (L.G.M.); meliquies@gmail.com (M.Q.); ljubica@unicamp.br (L.T.)

**Keywords:** metabolomics, antiseizure medication, ^1^H Nuclear Magnetic Resonance, focal epilepsy, response to treatment

## Abstract

A major challenge in the clinical management of patients with mesial temporal lobe epilepsy (MTLE) is identifying those who do not respond to antiseizure medication (ASM), allowing for the timely pursuit of alternative treatments such as epilepsy surgery. Here, we investigated changes in plasma metabolites as biomarkers of disease in patients with MTLE. Furthermore, we used the metabolomics data to gain insights into the mechanisms underlying MTLE and response to ASM. We performed an untargeted metabolomic method using magnetic resonance spectroscopy and multi- and univariate statistical analyses to compare data obtained from plasma samples of 28 patients with MTLE compared to 28 controls. The patients were further divided according to response to ASM for a supplementary and preliminary comparison: 20 patients were refractory to treatment, and eight were responsive to ASM. We only included patients using carbamazepine in combination with clobazam. We analyzed the group of patients and controls and found that the profiles of glucose (*p* = 0.01), saturated lipids (*p* = 0.0002), isoleucine (*p* = 0.0001), β-hydroxybutyrate (*p* = 0.0003), and proline (*p* = 0.02) were different in patients compared to controls (*p* < 0.05). In addition, we found some suggestive metabolites (without enough predictability) by multivariate analysis (VIP scores > 2), such as lipoproteins, lactate, glucose, unsaturated lipids, isoleucine, and proline, that might be relevant to the process of pharmacoresistance in the comparison between patients with refractory and responsive MTLE. The identified metabolites for the comparison between MTLE patients and controls were linked to different biological pathways related to cell-energy metabolism and pathways related to inflammatory processes and the modulation of neurotransmitter release and activity in MTLE. In conclusion, in addition to insights into the mechanisms underlying MTLE, our results suggest that plasma metabolites may be used as disease biomarkers. These findings warrant further studies exploring the clinical use of metabolites to assist in decision-making when treating patients with MTLE.

## 1. Introduction

Epilepsy is a chronic neurological disorder characterized by persistent and long-lasting hyperactivity of groups of neurons, increasing one’s propensity to develop epileptic seizures [1,2]. In mesial temporal lobe epilepsy (MTLE), the epileptogenic focus is localized in the medial structures of the temporal lobe, mainly in the hippocampus [3,4,5]. MTLE is the most common type of focal epilepsy in adults. About 67–89% of patients with MTLE with hippocampal sclerosis have drug-resistant epilepsy and do not respond to currently available antiseizure medications (ASMs) [6,7]. In this sense, it has been determined that patients who do not reach seizure freedom when taking two ASMs (in monotherapy or combination therapy) may be defined as drug-resistant [8]. In addition to being exposed to the seizures’ intrinsic deleterious effects, pharmacoresistant patients experience the severe side effects of the ASM, often prescribed as polytherapy and in high doses [9]. Therefore, a significant challenge in the clinical management of patients with MTLE is identifying those who do not respond to ASM, allowing for the pursuit of other types of treatment, such as epilepsy surgery [10,11]. Indeed, recent reports have highlighted the importance of identifying biomarkers for response to therapy in epilepsies, including MTLE [12]. Several hypotheses explain the mechanism of ASM resistance. The current consensus is that this is a multifactorial condition where gene–gene and gene–environment interactions play important roles [12]. 

Metabolomics based on Nuclear Magnetic Resonance (^1^H-NMR) is used for complex sample analysis, principally because of the high reproducibility of NMR data, little or no sample preparation, and sample integrity preservation, which is particularly relevant when evaluating rare clinical samples [13,14]. In addition, NMR-based metabolomics may identify changes in biological samples linked to morphological and biochemical alterations associated with disease, thus assisting in early and precise diagnosis. Furthermore, identifying metabolites associated with specific phenotypes can contribute to a better understanding of disease mechanisms [15]. 

This study investigated the feasibility of using plasma metabolites as disease and pharmacoresistance biomarkers in patients with MTLE. In addition, we used the metabolomics data generated to gain insights into the mechanisms underlying MTLE and response to ASM. 

## 2. Results

### 2.1. Characteristics of Study Population

We ascertained 28 patients with a mean age of 54 years (ranging from 26 to 70); eight patients were responsive to treatment with ASM, reaching seizure freedom when using the combination of CBZ + CLB, and 20 were classified as refractory to treatment with this therapy, as they did not acquire seizure freedom. The main clinical characteristics of the patients studied are reported in Table 1. We only included patients currently using CBZ + CLB. However, all patients considered here as pharmacoresistant have failed several other ASM regimens. None of the patients presented generalized seizures 24 h before blood collection. Most patients presented signs of hippocampal sclerosis on magnetic resonance image-decreased hippocampus volume in T1 images with increased signal in T2/FLAIR [3]. Age at onset of epilepsy varied from 1 to 30 years old. We also studied samples from 28 controls with a mean age of 49 years. These were adults between the ages of 29 and 63 years.

### 2.2. Metabolomic Analysis 

Overall, we identified 27 metabolites in the samples studied (Appendix B, Table A1). Additionally, we found different groups of biomolecules, such as lipids, in the plasma of patients and controls. These biomolecules corresponded to different saturations’ fatty acids; amino acids such as alanine, isoleucine, leucine, valine, and glutamine; and aromatic amino acids such as tyrosine, histidine, phenylalanine, lactate, and glucose (Figure 1). See Appendix B, Table A1, and Figure A1 for the chemical shift assignments.

The largest variation in the acquired NMR data directions was visualized using the PLS-DA method. The PLS-DA scores plot (Figure 2A) clusters data for the patients and controls and the first principal component (1), explaining 39.1% of the data variation. The PLS-DA model goodness of fit (R^2^) value was 0.83, with a prediction (Q^2^) of 0.23. The metabolites that discriminate between the groups were identified based on the variables’ importance in the projection score (VIP score) from the PLS-DA analysis. The first 15 VIPs were combined with univariate analysis such as t-test and fold change (FC) (Table 2). We also evaluated the AUC of the ROC curve of the most important features of the model to calculate their predictability power (Figure 3). As a result, we identified five significantly altered metabolites when comparing patients with controls: glucose, saturated lipids, isoleucine, β-hydroxybutyrate, and proline (Table 2 and Figure 2C).

Furthermore, we also built a model to compare the patients based on their pharmacological responses. In this sense, a PLS-DA model was developed for comparing refractory with responsive MTLE patients (Figure A3), displaying scores plot with clustering for the two groups and the first principal component (1), which explains 44.6% of the data variation.

In addition, a constructed model alone was not enough to consider the model prediction. However, despite not observing considerable predictability for our method in PLS-DA parameters, we proposed some candidate variables according to their PLS regression coefficients. Those were classified based on the variables’ relative VIP. These metabolites were lipoproteins, lactate, glucose, proline, isoleucine, and unsaturated lipids (Table 3 and Figure A3C). Nonetheless, these metabolites did not present significant differences by *t*-test when comparing patients with responsive and refractory MTLE.

A comparative evaluation of the metabolome between these two groups of patients, as presented here, could potentially reveal metabolic features related to the process of pharmacoresistance. However, as we did not obtain sufficient significance and predictive power in the statistical model, the information found for such a comparison, having lactate, glucose, proline, isoleucine, and unsaturated lipids only suggestively support their importance in such a condition. Thus, for a more assertive elucidation of the discriminating metabolites between patients with refractory and responsive MTLE, such an assessment should be pursued in a larger sample size.

Additionally, we performed pathway enrichment analysis with the differentially abundant metabolites to compare patients and controls. For such an analysis, we identified as main altered pathways: lactose degradation, glucose-alanine cycle, lactose synthesis, transfer of acetyl groups into mitochondria, glycolysis, gluconeogenesis, fatty-acid biosynthesis, galactose metabolism, sphingolipid metabolism, arginine, and proline metabolism, Warburg effect, valine, leucine, and isoleucine degradation pathway as main altered pathways in patients (Figure 4A).

We also analyzed the diseases to which the differently abundant metabolites found in the comparison between patients and controls have been previously linked. They are diabetes mellitus (non-insulin-dependent), persistent hyperinsulinemic hypoglycemia, pyruvate carboxylase deficiency, respiratory chain deficiencies, and long-chain 3-hydroxyacyl-CoA dehydrogenase (LCHAD) deficiency—for a complete overview of the diseases found to be linked to the differentially abundant metabolites, refer to Appendix B, Figure A2.

Finally, we built networks correlating metabolites with their putative-related genes using the Kyoto Encyclopedia of Genes and Genomes (KEGG) database. We found 18 genes associated with the metabolites capable of discriminating patients from controls (Figure 4B). Then, this set of genes was used to search the human phenotype ontology gene setlist and the Gene Set Enrichment Analysis (GSEA) database. We found genes related to elevated hepatic transaminase, energetic metabolism, amino-acid metabolism, and inflammatory processes. 

## 3. Discussion

We used ^1^HNMR-based metabolomics to investigate plasma samples of patients with MTLE and controls. In addition, we compared patients with different responses to ASM, i.e., those refractory and responsive to ASM. We searched for metabolic traits that could discriminate between the groups using a hypothesis-free design. Furthermore, we aimed to gain insights into the mechanisms underlying MTLE and response to treatment with ASM with the metabolomics data obtained. Because different epilepsy syndromes present different mechanisms, which may influence the biological processes leading to pharmacoresistance [12], we performed our study exclusively on patients with a single, well-defined epilepsy syndrome, MTLE. Furthermore, the use of ASM may lead to different metabolomics signatures [16]; thus, we selected patients using the same therapeutic regimen, CBZ + CLB, minimizing the source of bias. Moreover, the occurrence of seizures in proximity to the time of sample collection may temporarily affect the metabolic profile in plasma; hence, we only collected samples from patients who were seizure-free 24 h before sample collection. 

We found five altered metabolites comparing patients and controls: glucose, saturated lipids, 3-hydroxybutyrate, isoleucine, and proline. These biomolecules are related to metabolic processes involved mainly in energetic and amino-acid metabolisms, especially those related to the cell’s anaplerotic reactions. Noteworthy is the presence of lipids as one of the discriminating elements between patients and controls. 

In this study, we identified lower glucose levels in patients compared to controls. In this sense, changes in the energetic metabolism of neurons and glial cells have been frequently reported in neurological disorders. They are related to the preferential and intense usage of glucose by the nervous tissue. In addition, neurotransmitters, such as glutamate, acetylcholine, and gamma-aminobutyric acid, are dependent on energy metabolism [17]. The main molecule responsible for transferring and storing energy, the adenosine 5’-triphosphate (ATP), is produced through the glucose catabolism in the glycolysis, tricarboxylic acid (TCA) cycle, oxidative phosphorylation, and several other pathways related to the energetic metabolism, which were also present in the enriched pathways analysis performed in this study (Figure 4). Moreover, it is well-known that energy metabolism is indispensable for neurotransmitter production and release and the activity of ion channels. These are the essential players in the known mechanisms leading to epilepsy [18,19]. Therefore, an energetic failure due to abnormalities in glucose concentration may lead to increased seizure susceptibility [20]. Indeed, previous ^1^H-NMR studies revealed abnormal glucose concentrations in adult patients under different combinations of ASM [16] and in drug-free pediatric patients [21], presenting focal and generalized seizure types. Unfortunately, these studies presented important limitations since the diagnosis of epilepsy syndrome was not informed for both studies [16,21], and one of the studies included patients using different types of ASM [16]. Furthermore, none of the studies discussed above [16,21] informed if the patients presented seizures 24 h before sample collection, which may have introduced significant heterogeneity and bias toward identifying altered metabolites due to the ASM used and/or/or after a major seizure.

In addition, other energy sources become necessary in an energy-failure scenario since the decrease in ATP availability requires the recruitment of other biomolecules for energy production. In this context, lipidic support is crucial to cell maintenance [22]. In our study, some lipids corresponding to saturated fatty acids have been found elevated in patients and can be related to altered energetic metabolism since this group of molecules, abundantly found in lipoproteins, such as VLDL, HDL, and LDL, can be constantly displaced to many different tissues to supply the energy input [23].

Other important energy sources for neural cells are gluconeogenesis, glycogenolysis, and beta-oxidation. The ketone bodies can also perform this role, being synthesized depending on the metabolic disposition of the cell in prolonged periods of energetical deficit [24]. These molecules—acetoacetate, acetone, and β-hydroxybutyrate—contribute mainly to the brain-tissue energy input, as they can cross the blood–brain barrier, supplying about 60% of the central nervous system (CNS) [25]. Our study found increased levels of β-hydroxybutyrate in patients, even though signals of saturated lipids overlay the chemical shift identified for this molecule. This finding also indicates a state of energetic failure in the cells of patients with MTLE. 

In addition to the well-known therapeutic strategy of the ketogenic diet in epilepsy, diets rich in BCAA have been shown to have positive effects on seizure control in an animal model [26] and combined with the ketogenic diet in children with epilepsy [27]. However, a recent study failed to demonstrate that chronic ingestion of BCAA is effective in the long-term reduction of seizures in an animal model of MTLE [28]. It has been proposed that an imbalance in the concentrations of amino acids such as valine, leucine, and isoleucine may lead to increased glutamate levels in the nervous tissue, and consequently, to increased hyperexcitability and excitotoxicity, resulting in neuroinflammation, which is known to occur in epilepsy [27,29,30]. Interestingly, we found increased levels of isoleucine in patients compared to controls, indicating a complex relationship between isoleucine concentration in MTLE and response to ASM treatment. 

Furthermore, we found increased proline levels in patients compared to controls. This nonpolar amino acid has been linked to seizures and cognitive dysfunction [31], as well as to *PYCR2* gene expression linked to microcephaly and hypomethylation [32], along with the *PARS2 gene* related to infantile-onset encephalopathy [33]. Furthermore, isoleucine is a metabolite linked to genes involved in neurological disorders, such as dementia and seizures: *BCAT1*, *BCAT2*, and *IARS2* (26980008 and 30098844).

Interestingly, we found that the metabolic profile of patients with MTLE was also linked to non-insulin-dependent diabetes mellitus (Mody). This association has been reported previously, and it may be disease-induced or linked to the chronic use of ASM [34,35,36]. Indeed, a disruption of glucose metabolism in epilepsy leading to the unusual use of glucose as a source of energy [20,37,38] and pro-inflammatory processes occurring in epilepsy [38,39,40] may be risk factors for the development of diabetes in patients with MTLE.

However, the influence of pharmacological therapy on the group of patients cannot be discarded. This set of alterations, mainly linked to the amino acids and lipidic metabolism pathways found in the comparison between patients and controls, could be related to the ASM use, not exclusively the disease hallmarks. Despite the low evidence of metabolic alterations in humans under the use of the ASMs addressed in this study, assessments on the influence of CBZ in animal models have already been reported. In this context, alterations in amino acids and carnitine metabolisms (related to the lipid metabolism) in models of exposure to different doses of CBZ using Zebrafish [41] and in a bivalve mollusc *Mytilus galloprovincialis* [42] were already described.

Overall, our study found that the levels of metabolites such as glucose, saturated lipids, β-hydroxybutyrate, isoleucine, and proline, can distinguish patients with MTLE from controls. In addition, the computational integration of the data obtained in the metabolomic analysis suggests that pathways related to energetic metabolism, excitatory neurotransmission, and inflammatory processes are abnormally regulated in patients with MTLE. Notably, recent studies using tissue from animal models of MTLE reported similar altered biological pathways and processes [38,39,40]. Finally, we found that isoleucine and proline, amino acids present in abnormal levels in patients, may be involved in the increased susceptibility to seizures.

### Strengths and Limitations 

Strengths of this study include a well-characterized and homogeneous cohort since only patients with confirmed MTLE using the same type of ASM were included in the study. In addition, all patients were followed prospectively by the same group of neurologists in an epilepsy clinic at a university hospital. In addition, we only included patients without generalized seizures in the last 24 h of blood collection. The diagnostic criteria used to identify patients with MTLE and characterize the response to treatment with ASM followed the recommendations by the International League Against Epilepsy [8,43]. Furthermore, we used an agnostic approach to identify metabolites in the patients’ plasma and controls through a robust, highly reproducible analytical technique. To our knowledge, this is the first study to comply with all the elements listed above. 

Our study also presents some limitations. First, because we recruited such a homogenous group of patients and controls, our study counted on a limited number of investigated individuals. Besides that, the access to these patients’ biofluids depends on their active follow-up in the epilepsy clinic, which increases the difficulty of obtaining them. This occurs mainly with the responsive patients, since they do not often return to the hospital due to their positive response to prescribed ASMs. Therefore, this reduced number of recruited individuals may have impacted the weak significance and predictability observed in our findings in comparing patients with responsive and refractory MTLE. 

Another important limitation in our study is the inherent influence of the ASMs on the metabolic profile when comparing patients to controls. Thus, the correlation between some of the metabolites identified may not be due to the disease but to the recurrent use of the ASMs well.

## 4. Conclusions

Overall, plasma levels of glucose, saturated lipids, β-hydroxybutyrate, isoleucine, and proline can distinguish patients with MTLE from controls. Most interestingly, glucose levels, isoleucine, proline, lipoproteins, unsaturated lipids, and lactate are different in patients with pharmacoresistant MTLE compared to responsive MTLE. In addition, the computational integration of the data obtained in the metabolomic analysis suggests that pathways related to energetic metabolism, excitatory neurotransmission, and inflammatory processes are abnormally regulated in patients with MTLE. The identified metabolites were linked to biological pathways related to cell-energy metabolism, inflammatory processes, and modulation of neurotransmitter release and activity, which can potentially contribute to the mechanisms underlying MTLE. Thus, in addition to insights into the mechanisms underlying MTLE, we provided a supplementary analysis comparing patients with refractory and responsive MTLE, showing suggestive evidence that plasma metabolites may be used as biomarkers of response to ASM in patients with MTLE. These findings warrant further studies exploring the clinical use of metabolites to assist in medical decisions in treating patients with MTLE. 

## 5. Materials and Methods

### 5.1. Study Population 

We studied 28 patients with MTLE at the University of Campinas (UNICAMP) hospital’s outpatient epilepsy clinic. All patients had confirmed MTLE diagnoses according to International League Against Epilepsy (ILAE) criteria [43] and were investigated using a standard protocol, including serial interictal electroencephalogram high-resolution 3 Tesla magnetic resonance imaging [44,45]. We only included adults between the ages of 18 and 65. Patients were classified into two groups according to the patients’ responses to the treatment with ASM refractory or responsive following ILAE recommendations [8]. In addition, we only included patients using carbamazepine (CBZ) with clobazam (CLB), the therapy most frequently used by our epilepsy clinic patients. None presented generalized seizures 24 h before blood collection. Furthermore, we excluded patients using drugs for secondary clinical conditions that could have interacted with the ASM. We also studied samples from 28 control individuals. These were adults between 18 and 65 who were not in treatment for any neurological or chronic clinical conditions. We also excluded individuals under therapy with medications that could alter the function of key metabolism enzymes, such as cytochrome P450 isoforms. 

### 5.2. Blood Collection and Plasma Preparation

Peripheral blood (4 mL) was collected in standard EDTA tubes (Vacutainer; Becton Dickinson, Franklin Lakes, NJ, USA). Blood samples were kept on ice for up to two hours until the plasma was separated and centrifuged at 2500 rpm for 10 min at 4 °C. Then, the obtained plasma was aliquoted and stored at −80 °C until further analysis.

### 5.3. ^1^H-NMR Spectroscopy Analyses

Samples were thawed in an ice bath (4 °C), diluted 1:1 (*v*/*v*) in deuterium oxide (D_2_O, 99.9% with 0.03% of trimethylsilyl propanoic acid, TSP, from Sigma Aldrich, St. Louis, MO, USA), and transferred into 5 mm NMR tubes. A Bruker AVANCE III 600 spectrometer (Bruker Biospin, Karlsruhe, Germany), equipped with a TBI (Triple Resonance Broadband Inverse) probe, was used. All ^1^H-NMR spectra were acquired at 25 °C, for example, Carr–Purcell–Meiboom–Gill (CPMG) pulse sequence (cpmgpr1d), with 128 scans, was explored further. The two-dimensional (2D) total correlation spectroscopy (TOCSY) experiments were recorded with mlevgpphw5 pulse sequence and 300 scans of randomly selected samples. 

### 5.4. Data Analysis: NMR Data Processing 

The obtained spectra were processed and standardized using MestreNova software (MestrelabResearch S.L.). First, the spectra were referenced using TSP (0.0 ppm), then aligned, and their phase was corrected. Next, peaks were normalized by the total area, and the matrix was prepared using spectral bins (0.005 ppm). A normalization with a constant sum (100) of the entire spectrum intensity was employed, reducing the differences in concentration between the plasma samples. In addition, that, water and EDTA spectral regions were removed.

Such a data matrix was processed using MetaboAnalyst 5.0 platform (Xia, McGill University) [46]. Data filtering, normalization by the sum, and Pareto scaling (mean-centered and divided by the square root of the standard deviation of each variable) were used in data preprocessing before the statistical tests, and 56 samples against 1620 variables (bins) for ^1^H-NMR CPMG were analyzed.

Furthermore, we performed multivariate principal component analysis (PCA), partial least-squares discriminant analysis (PLS-DA), orthogonal (o) PLS-DA, leave-one-out cross-validation (LOOCV), and variable importance in projection (VIP) scores. The first fifteen VIPs were correlated with univariate analyses, such as fold change (FC) and the *t*-test (*p*-value < 0.05). The area under the curve (AUC) of the receiver operator characteristics (ROC) curve of the selected features was also used to evaluate their prediction power by using the MetaboAnalyst 5.0 platform (Xia, McGill University) [46]. The chemical compounds were identified with the support of the Human Metabolome Database (HMDB) h [47] and Biological Magnetic Resonance Data Bank (BMRB) [48]. Metabolic and disease pathways analyses were performed using the Metaboanalyst function Metabolic Set Enrichment Analysis (MSEA) with an over-representation analysis (ORA) algorithm. We also built a network of compound-gene relations using the MetScape app from Cytoscape [49] to identify enriched pathways and visualize changes in metabolite data. All the above enrichment analyses were generated using the Kyoto Encyclopedia of Genes and Genomes (KEGG) [50] and the Small Molecule Pathway Database (SMPDB) [51].

## Figures and Tables

**Figure 1 metabolites-12-00446-f001:**
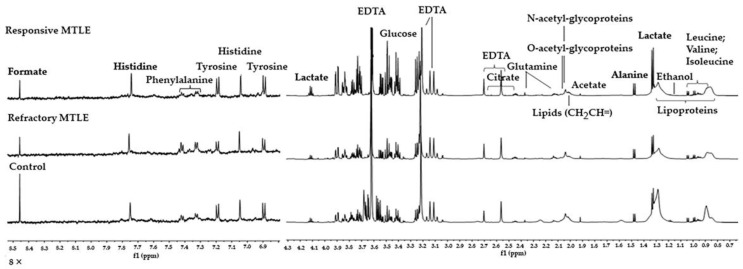
The representative ^1^H-NMR spectra of plasma samples of control, responsive, and refractory MTLE subjects. Spectral regions from 0.5 to 9.0 ppm, acquired using CPMG (cpmgpr1d) pulse sequence, are shown. The 7.0 to 8.5 ppm regions were zoomed in (8×) for better visualization. The regions of D_2_O and EDTA were removed (//). The following metabolites were identified: lipoproteins; leucine, valine; isoleucine; lactate; alanine; acetate; *N*-acetyl-glycoproteins; *O*-acetyl-glycoproteins; glutamine; glucose; tyrosine; histidine; phenylalanine; formate.

**Figure 2 metabolites-12-00446-f002:**
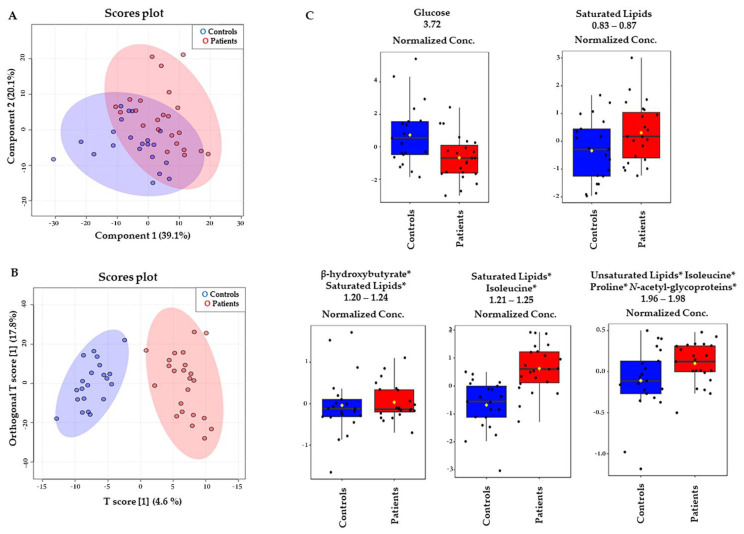
Multivariate analysis of ^1^H-NMR (CPMG) plasma spectra. (**A**) PLS-DA results with the accuracy of 76%, R2 0.83, and Q2 0.23. (**B**) O-PLS-DA results. The information about response to treatment with ASM in patients with MTLE was not implemented into the models. (**C**) Box plots representing the variations of the relative concentrations (measured as peak intensities) of metabolites whose VIP scores > 2 according to PLS-DA results. The black dots represent the metabolite levels in all samples, and the yellow diamond represents the average value. Patients with MTLE (red) and controls (blue). * = overlaid signals.

**Figure 3 metabolites-12-00446-f003:**
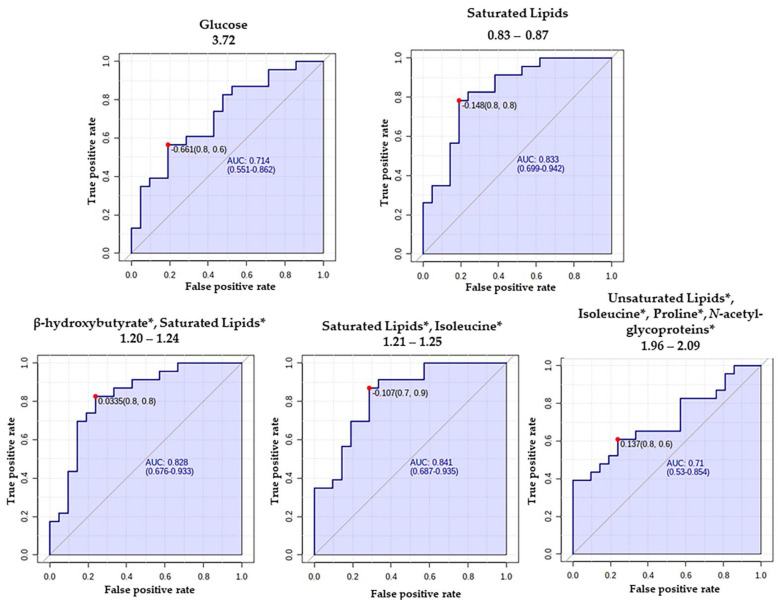
Receiver operator characteristics (ROC) curves and their respective areas under the curve (AUC) were calculated for the most important features determined by VIP values to compare MTLE patients and controls. * = overlaid signals.

**Figure 4 metabolites-12-00446-f004:**
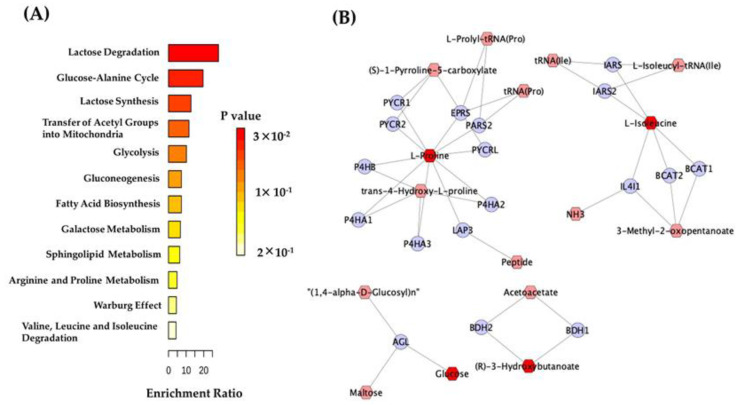
Metabolic pathways were identified by the Metabolite Set Enrichment Analysis (MSEA) of the discriminant metabolites that were identified when comparing (**A**) patients and controls. (**B**) Network compound genes built with the discriminant metabolites comparing patients and controls. Blue circles—annotated genes; pink hexagon—metabolite’s ligands; red hexagon—discriminant metabolites present in the KEGG database.

**Table 1 metabolites-12-00446-t001:** The main characteristics of the 28 patients with MTLE included in the study. All patients were using the ASM combination, CBZ + CLB, and most of them presented signs of hippocampal sclerosis on magnetic resonance imaging—decreased hippocampus volume in T1 images with increased signal in T2/FLAIR images.

ID	Sex	Age (Years)	Age at Onset of Epilepsy	Hippocampal Abnormalities	Group	Response to Treatment with ASM Combination, CBZ + CLB
1	F	63	15	LHS	MTLE	Refractory
2	M	60	26	Bilateral	MTLE	Refractory
3	F	57	1	LHS	MTLE	Refractory
4	M	59	15	LHS	MTLE	Refractory
5	F	70	17	LHS	MTLE	Refractory
6	M	58	14	RHS	MTLE	Refractory
7	F	50	2	LHS	MTLE	Refractory
8	M	50	19	LHS	MTLE	Refractory
9	F	37	7	LHS	MTLE	Refractory
10	M	26	5	LHS	MTLE	Refractory
11	M	54	10	LHS	MTLE	Refractory
12	F	60	16	RHS	MTLE	Refractory
13	F	26	7	RHS	MTLE	Refractory
14	F	62	23	Bilateral	MTLE	Refractory
15	M	60	20	RHS	MTLE	Refractory
16	M	62	14	LHS	MTLE	Refractory
17	F	61	2	RHS	MTLE	Refractory
18	M	46	1	LHS	MTLE	Refractory
19	F	54	8	LHS	MTLE	Refractory
20	F	51	30	None	MTLE	Refractory
21	F	43	7	LHS	MTLE	Responsive
22	M	45	8	RHS	MTLE	Responsive
23	F	65	3	LHS	MTLE	Responsive
24	M	55	31	RHS	MTLE	Responsive
25	F	58	17	RHS	MTLE	Responsive
26	M	47	19	LHS	MTLE	Responsive
27	F	56	20	LHS	MTLE	Responsive
28	F	70	18	LHS	MTLE	Responsive

MTLE: mesial temporal lobe epilepsy; ID: patient identification; sex: male/female (M/F); age: the age at investigation. LHS: left hippocampal sclerosis; RHA: right hippocampal sclerosis; bilateral: bilateral hippocampal sclerosis.

**Table 2 metabolites-12-00446-t002:** Table showing the metabolites identified in different concentrations and their respective chemical shifts elucidated by the highest VIP values (VIP score). The *p*-values, calculated from the t-test, FC (fold change), and false discovery rate (FDR) are also shown.

MTLE versus Control—VIP Score					
Metabolites	Chemical Shift (Multip; Assign.)	Vip Score	*p*-Value	FC	FDR
Glucose	3.68–3.78 (m, CH)	5.12	6.0000 × 10^−3^	1.30	0.196
Saturated Lipids	0.83–0.87 (m, CH_3_)	5.02	0.0918 × 10^−3^	0.76	0.023
Saturated Lipids, Isoleucine *	1.21–1.25 (m, -CH_2_-)	3.80	0.0817 × 10^−5^	0.82	0.023
β-Hydroxybutyrate, Saturated Lipids *	1.20–1.24 (m, -CH_2_-	3.15	0.129 × 10^−3^	0.82	0.027
Unsaturated lipids, Isoleucine, Proline, and *N*-Acetyl-glycoproteins *	1.96–2.09 (m, -CH_2_-CH=)	2.16	0.0905 × 10^−3^	0.85	0.198

Multip = multiplicity, where s (singlet), d (doublet), t (triplet), dd (doublet of doublets), m (multiplet), l (broad); Assign. = assignment of these signals; * = overlaid signals.

**Table 3 metabolites-12-00446-t003:** List of metabolites identified in different concentrations of refractory and responsive MTLE and their respective chemical shifts elucidated by the highest VIP values (VIP score). The *p*-values, calculated from the *t*-test, FC (fold change) are also shown.

Refractory MTLE versus Responsive MTLE—VIP Scores				
Metabolites	Chemical Shift (Multip.; Assign.)	VIP Score	*p*-Value	FC
Lipoproteins	1.28 (m, CH)	6.66	0.05	1.209
Lactate	1.33 (d, CH_3_)	5.41	0.05	1.159
Glucose	3.41 (m, CH_2_)	4.81	0.05	0.752
Exclusively unsaturated lipid	2.06 (l, CH_2_CH=)	1.71	0.05	0.857
Isoleucine	0.94 (t, CH_3_)	1.57	0.05	0.886
Proline	3.36 (m, CH)	1.13	0.05	0.658

Multip = multiplicity, s (singlet), d (doublet), t (triplet), dd (doublet of doublet), m (multiplet), l (broad); Assign. = assignment of these signals.

## Data Availability

The data analyzed in this study has been presented in the manuscript and in the Appendix A.

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
