# Peer review of "Circulating Metabolites as Biomarkers of Disease in Patients with Mesial Temporal Lobe Epilepsy"

_metabolites, 2022, doi:10.3390/metabo12050446_

Round 1

Reviewer 1 Report

The authors present a metabolomics study on plasma samples using 1H NMR spectroscopy followed by uni- and multivariate statistical analysis. In particular, they looked for biomarkers related to patients with mesial temporal lobe epilepsy who fail to respond to antiseizure medication. They did indeed confirm  the identity of several biomarkers and suggest further investigations to aid in decision making for disease treatment. Overall, the study is well conceived and executed and represents the maturity 1H NMR spectroscopy in probing metabolomic profiles for defined clinical problems. I only have minor suggestions for improving:

Please state the limitations of your study. In particular, you only use 28 patients and 28 controls. How might this number of patients impact your findings?

Secondly, the fonts for the figures are so small and very difficult to read. In particular, Figure 4 is nearly unreadable, yet these metabolic pathways are identified from the data. Please increase font size for figures, and especially be sure Figure 4 is readable.

Author Response

-We complemented the topic "Strengths and limitations" following the reviewer's suggestions: on page 12, lines 46-51; and on page 13, lines 1-3.

-We have replaced and corrected the fonts according to the reviewer's suggestions. 

All changes in the manuscript are marked in yellow. 

Reviewer 2 Report

In this paper the authors applied an NMR-based metabolomics strategy for the investigation of plasma samples of patients with epilepsy. The aim of the project was twofold: discriminating patients from healthy controls and discriminating responders from refractory patients. The paper has some strengths: a well characterized cohorts with a uniform therapy, strict diagnostic criteria to identify responders and non-responders, state of the art NMR methods. However, there are also major limitations that prevent the publication in the present form:
 1) The number of subjects involved is quite limited: 28 patients (20 refractory, 8 responders) and 28 controls. This severely questions the impact of the paper and the significance of the results, considering also the weak discrimination observed.
2) For the comparison of controls vs patients, the authors observed both a significant discrimination in the PLS (73% accuracy), and some statistically discriminant metabolites. This is not the case for the comparison of refractory vs responder patients. However the authors stated: “Multiple comparison correction was performed through the false discovery rate (FDR) at a level of 0.05, but it did not result in any significant metabolite (p.adjust). Therefore, we considered statistically significant results for which p-values were < 0.1”. I doubt that this is a correct way of proceeding. The significance threshold must be fixed before the analysis (based on the number of subjects and an estimation of the observed effects; without any prior knowledge, the usual 0.05 level is an accepted threshold that avoids false discoveries).
3) I do not consider a problem the reporting of negative results. So, the negative results for the comparison of refractory vs responder patients can be maintained in the manuscript. However, the pathway analysis (performed on the basis of 6 not-significant metabolites coming from a comparison of 20 vs 8 patients) and the further discussion involving them are severely questionable. I suggest removing from the manuscript this pathway analysis and any discussion of metabolites and pathways for this comparison.
4) Because all patients are on treatment (probably since many years), the authors cannot exclude that the discrimination observed between controls and patients is just an effect of the therapy, not of the disease. This needs to be assessed and discussed.    

Author Response

  • We agree, in part, with the reviewer that the reduced number of recruited individuals may have impacted the weak significance and predictability observed in the comparison between responsive and refractory patients. But not when comparing patients and controls. We have discussed the issue further in the manuscript and toned down the discussion of the results for the less significant results obtained in comparing responsive and refractory patients: page 12, lines 46-51; and page 13, lines 1-7.
  • We agree with the reviewer, and we have modified the manuscript according to the reviewer's suggestions, we added the values of FDR to Table 2 and changed the segment that described the statistical approach used: page 3, lines 32-34.
  • We followed the reviewer's suggestions and modified the manuscript to reflect that the results of the comparison between the two groups of patients are not significant. Also, we removed all the enriched pathway analyses and the subsequent discussions of the metabolites found for the comparison between patients. Changes are marked in yellow in the results and discussion. 
  • We added a discussion about the reviewer’s observation: on page 12, lines 14-23, and on "strengths and limitations" on page 13, lines 4-7.

Reviewer 3 Report

Overall, this paper was novel and fascinating. The manuscript was very well written, organized, and easy to follow. The experimental methods and study designs were appropriate. The result sections were interesting for me, especially regarding the beta-hydroxybutyrate portion. The only part that made me scratch my head was the 'strengths and limitations' portion since it did not seem to list the actual limitations of the study. But other than that, I did not find any issue or concern. So great job to all the authors who have devoted their time and effort to this research project.

Author Response

We have modified the statement about limitations following the reviewer's suggestion: page 12, lines 46-51; and page 13, lines 1-7.

Changes are marked in yellow in the revised manuscript. 

Reviewer 4 Report

The authors used untargeted NMR-based metabolomics to characterize patients with mesial temporal lobe epilepsy (MTLE) and the response to antiseizure medication (ASM). The analysis was based on EDTA plasma samples. Multivariate and univariate statistical were used to analyse data obtained from 21 samples of 28 patients with MTLE and 28 controls. Moreover, the authors classified the patients according to response their response to ASM treatment and performed characterization of the metabolites and metabolic pathways involved in the responsiveness.

For the metabolomic characterization of MTLE profile, the authors reported multivariate models able to discriminate between patients and controls. The authors also compared the group of patients with controls with univariate analysis and found that some metabolites were significantly different between the two groups. For this part, I have not major concerns but I strongly suggest to confirm the differences in metabolite levels in the two groups by calculating integrals of each metabolite specific signal areas (or signal deconvolution). The used of bucket areas can lead to misleading results because the signals of more metabolites can fall into the same bucket.   

Contrarily I have major concerns about the comparison between responder and non-responder subjects. Regarding this point, the authors reported a PLS-DA model with a R2 value of 0.13 and a Q2 value of -0.19. A permutation test should be used to assess the significance of the PLS-DA models. They also reported that lipoproteins (p = 0.05), lactate (p = 0.05), glucose (p = 0.05), unsaturated lipids (p = 0.05), isoleucine (p = 0.05), and proline (p = 0.05) discriminate between the two groups of patients classified according to response to ASM and proposed that pyruvate metabolism may be linked to resistance to ASM. All the reported results for the comparison between responders and non-responders are not statistically significant. Thus, for my point of view all the discussion regarding this point is not supported by the results and the proposed involved pathways are not reliable.

Author Response

-Thank you for the suggestions. We have confirmed the differences in the main metabolite levels by calculating the corresponding integral data of their picks. We also added the t-test and the FDR values in Table 2 and constructed the area under the curve (AUC) of the receiver operator characteristics (ROC) curve of the selected features to evaluate metabolites' prediction power. This new information was added to the Results, page 8.

-We agree that the predictability values for comparing the two groups of patients, responders and non-responders, did not reach a considerable predictive level. However, we decided to maintain and rewrite the results for such a comparison, seeking to classify them as preliminary with suggestive evidence for a role in pharmacoresistance, which should be analyzed in a larger sample size. Therefore, we have removed the analysis of enriched pathways and the consecutive discussions for such comparison from the manuscript.

-All changes in the manuscript are marked in yellow

Round 2

Reviewer 2 Report

The paper is improved. No further comments from my side.

Reviewer 4 Report

The authors have taken into account all the points. I have not further comments.